# Differential transcript usage in the Parkinson's disease brain

**Fiona Dick** [1,2], **Gonzalo S. Nido** [1,2], **Guido Werner Alves** [3,4], **Ole-Bjørn Tysnes** [1,2], **Gry Hilde Nilsen** [1,2], **Christian Dölle** [1,2], **Charalampos Tzoulis** [1,2]*

**1** Neuro-SysMed, Department of Neurology, Haukeland University Hospital, Bergen, Norway, **2** Department of Clinical Medicine, University of Bergen, Bergen, Norway, **3** The Norwegian Center for Movement Disorders and Department of Neurology, Stavanger University Hospital, Stavanger, Norway, **4** Department of Mathematics and Natural Sciences, University of Stavanger, Stavanger, Norway

* charalampos.tzoulis@uib.no

**Data Availability Statement:** The datasets supporting the conclusions of this article are included within the article and its supplementary files. The source code and raw data to reproduce the results of the analyses is available in the GitHub

## Abstract

Studies of differential gene expression have identified several molecular signatures and pathways associated with Parkinson's disease (PD). The role of isoform switches and differential transcript usage (DTU) remains, however, unexplored. Here, we report the first genome-wide study of DTU in PD. We performed RNA sequencing following ribosomal RNA depletion in prefrontal cortex samples of 49 individuals from two independent case-control cohorts. DTU was assessed using two transcript-count based approaches, implemented in the DRIMSeq and DEXSeq tools. Multiple PD-associated DTU events were detected in each cohort, of which 23 DTU events in 19 genes replicated across both patient cohorts. For several of these, including *THEM5*, *SLC16A1* and *BCHE*, DTU was predicted to have substantial functional consequences, such as altered subcellular localization or switching to non-protein coding isoforms. Furthermore, genes with PD-associated DTU were enriched in functional pathways previously linked to PD, including reactive oxygen species generation and protein homeostasis. Importantly, the vast majority of genes exhibiting DTU were not differentially expressed at the gene-level and were therefore not identified by conventional differential gene expression analysis. Our findings provide the first insight into the DTU landscape of PD and identify novel disease-associated genes. Moreover, we show that DTU may have important functional consequences in the PD brain, since it is predicted to alter the functional composition of the proteome. Based on these results, we propose that DTU analysis is an essential complement to differential gene expression studies in order to provide a more accurate and complete picture of disease-associated transcriptomic alterations.

## Author summary

Altered expression has been found at the level of genes and pathways in the brain of individuals with Parkinson's disease but remains unexplored at the level of individual transcripts. Thus, it is largely unknown whether transcript-specific events, for instance due to altered splicing or post-transcriptional modifications, occur in the Parkinson's disease

repository "DTUinPDbrain", https://github.com/fifdick/DTUinPDbrain under the GPL public license v3.0. Result tables, raw counts (in TPM) and sample information used to run the analyses are available at Figshare: https://figshare.com/articles/dataset/Differential_transcript_usage_in_the_Parkinson_s_disease_brain/12941945 Raw FASTA files include sensitive information and can not be shared publicly.

**Funding:** This work is supported by grants from The Research Council of Norway (288164, ES633272) (https://www.forskningsradet.no/en/) and Bergen Research Foundation (BFS2017REK05) (https://mohnfoundation.no/engelsk-rekruttering/?lang=en). Both of these were received by CT. The funders had no role in study design, data collection and analysis, decision to publish, or preparation of the manuscript.

**Competing interests:** The authors have declared that no competing interests exist.

brain. Using RNA sequencing data from 49 brain samples, we performed a transcriptome-wide study of differential transcript usage in Parkinson's disease. We identified transcript-specific changes in multiple genes, and many of these were predicted to have important functional consequences on the encoded protein, such as altered subcellular localization or total protein levels. Interestingly, the vast majority of these transcript-specific changes were not detected by conventional differential gene expression analysis. Our findings suggest that analyses of differential transcript usage can provide additional insight into the transcriptomic landscape of complex brain disorders.

## Introduction

Parkinson's disease (PD) is the second most prevalent neurodegenerative disorder, affecting more than 1% of the population above the age of 60 years [1]. Both genetic and environmental factors influence the risk of PD, but the molecular mechanisms underlying disease initiation and progression remain unknown. Studies of differential gene expression (DGE) employing microarrays or RNA sequencing (RNA-Seq) have identified molecular signatures associated with PD, including various aspects of mitochondrial function, protein degradation, neuroinflammation, vesicular transport and synaptic transmission [2].

An important limitation of DGE studies, however, is that they do not account for isoform diversity. Most genes encode more than one transcript isoform (henceforth called isoform), arising from alternative splicing, alternative usage of transcription start sites, or post-transcriptional regulation events such as alternative cleavage and polyadenylation [3]. Distinguishing between isoforms is essential, as these can encode proteins with different functions and/or subcellular localizations, or no protein product at all. Isoforms can also be associated with varying degrees of mRNA stability, for example by varying the length of the 3'-untranslated regions, which ultimately influences the rate of translation and hence the quantity of the encoded protein [4]. Moreover, differential splicing can impact cellular function without causing major changes on the levels of expressed protein. The diversity of tissue-specific isoform expression patterns is mainly attributed to differential usage of untranslated transcripts and/or non-principal isoforms, suggesting that even small changes in isoform usage can have a substantial effect on the composition and function of the proteome [5].

An efficient method to characterize differences in the isoform landscape is via differential transcript usage (DTU) analysis. DTU is a measure of the relative contribution of one transcript to the overall expression of the gene (i.e. the total transcriptional output). The analysis is based on individual transcript read counts normalized to the sum of all transcript read counts of the gene. This sets DTU apart from differential transcript expression (DTE), where the individual transcript counts are investigated independently from the context of the total transcriptional output. DTU requires at least one DTE event for the usage ratio between the transcripts of a gene to change. In contrast, DTE can occur without DTU, when the expression of an isoform is altered but its relative contribution to the total transcriptional output remains unchanged [6].

Individual transcript-level information—DTE or DTU—is lost in conventional DGE analysis, where the counts of individual transcripts are collapsed at the gene level. DTU events changing in opposite directions (e.g. when one transcript is up-regulated and another down-regulated) may cancel out at the gene level. Thus, transcript usage quantification has the potential to identify candidate genes and processes which would otherwise remain concealed in traditional DGE and DTE studies.

In the human brain, specific transcript usage profiles have been associated with neuronal development and aging [7] as well as with disease [8], including neurodegeneration [9, 10]. Current evidence suggests that differential splicing and DTU may be implicated in PD [11]. Disease-associated alternative splicing has been reported for genes linked to idiopathic and monogenic PD, including *SNCA* [12], *PRKN* [12, 13] and *PARK7* [14]. With the exception of these targeted, hypothesis-based studies, however, the role of DTU in PD remains largely unexplored and no genome-wide DTU studies have been carried out to date.

In the present study we report the first genome-wide analysis of DTU in PD. We show that DTU does occur in the PD brain and identify genes that show robust, altered isoform ratios across two separate cohorts of individuals with idiopathic PD and neurologically healthy controls: a discovery cohort from the Park West study [15] ($n = 28$) and a replication cohort from the Netherlands Brain Bank ($n = 21$).

## Results

### Multiple DTU events are detected in the PD prefrontal cortex

We first analyzed RNA-Seq data from the prefrontal cortex of our discovery cohort ($n = 17/11$ PD/controls; Table A in S1 File), using two alternative approaches (DRIMSeq [16] and DEXSeq [17]) to characterize DTU between PD and controls. Statistically significant DTU surviving multiple testing correction are referred to as DTU events and a gene exhibiting at least one DTU event is referred to as a DTU gene (detailed definitions are provided in the Methods).

In the discovery cohort, DTU analysis was based on $n = 40, 520$ transcripts and identified 814 DTU events in 584 DTU genes. The analysis with DEXSeq identified 254 DTU genes and 495 DTU genes were reported by DRIMSeq, with 165 detected by both methods (Fig 1A). The number of single DTU events per DTU gene ranged from one to three (Table 1). The most common Ensembl transcript biotype involved in DTU events was "protein coding" for both DEXSeq and DRIMSeq, followed by "processed transcript" (i.e., transcripts not containing an ORF) and "retained intron" (i.e., transcripts containing intronic sequences) (Fig 1B). We tested for overrepresentation of DTU events across transcript biotypes using Fisher's exact test and found that DTU events were overrepresented in 3 categories for DRIMSeq after multiple testing correction at alpha 0.05 (protein coding, retained intron, antisense). Although no categories were significantly overrepresented after Bonferroni correction using DEXSeq, the lowest p-values were for "antisense" and "protein coding", in agreement with DRIMSeq. Test statistics for each of the biotype categories are listed in S1 Table.

Visualization of the overall behavior of the effect size as a function of the mean transcript expression (MA-plot) and nominal transcript significance (Volcano-plot) are shown in S1A and S1B Fig. The p-value distribution varied depending on the number of transcripts a gene possessed. This variation behaved differently in DRIMSeq and DEXSeq—the p-value distribution became more uneven with increasing numbers of transcripts in DRIMSeq and decreasing number of transcripts in DEXSeq (S2C Fig). A list of identified DTU events is provided in Table B in S1 File.

Gene-set enrichment analysis (GSEA) of the DTU genes showed clusters of enriched pathways related to regulation of cell development, identical protein binding and perinuclear region of cytoplasm as the top most significant in each of the GO Ontology categories (Biological process, Molecular function, Cellular component) (Table 2).

To validate our methodology, we sought to confirm relative transcript abundances of genes with a DTU event by quantitative PCR (qPCR). To this end, we selected two genes fulfilling the following criteria: i) adequate individual transcript expression levels (i.e., the transcript

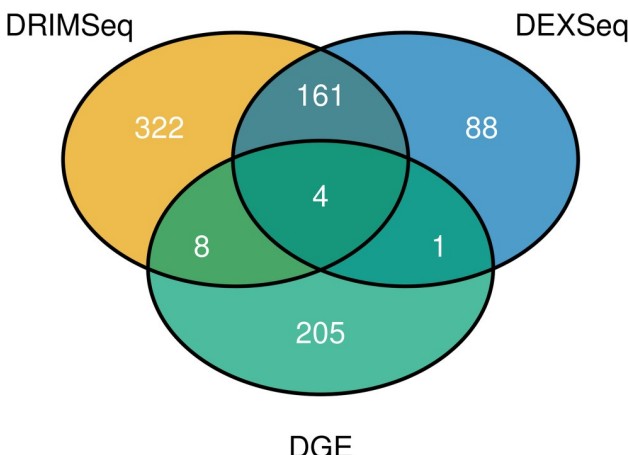

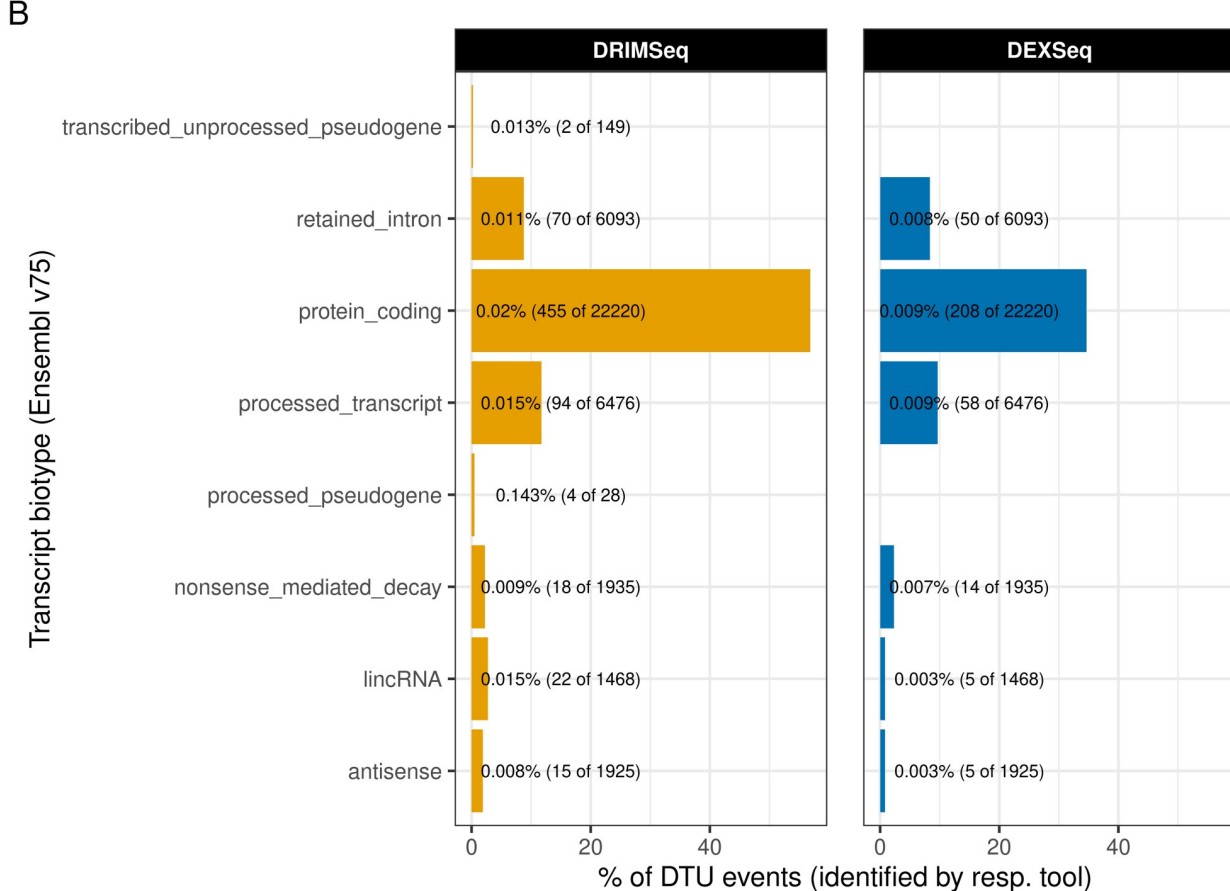

**Fig 1. Overlap of DTU genes and transcripts between DEXSeq and DRIMSeq.** A: Venn diagram showing the overlap between DTU genes resulting from analyses using DEXSeq and DRIMSeq, and genes that show DGE in the discovery cohort. B: Distribution of DTU events across defined transcript biotypes for each of the two tools (panels). Transcript biotypes are arranged on the y-axis, with the percentage of DTU events in each biotype category of all tool-specific DTU events represented on the x-axis. Text labels show the percentage of DTU events relative to the number of transcripts tested in each biotype category.

**Table 1. Distribution of the number of DTU events per gene.**

| Tool | 1 transcript | 2 transcripts | 3 transcripts |
|---|---|---|---|
| DEXSeq | 173 | 76 | 5 |
| DRIMSeq | 312 | 181 | 2 |

was present in both cohorts after pre-filtering and detectable by qPCR) and ii) sufficiently distinct exonic composition of the individual transcripts to allow transcript-specific amplification (i.e., it was possible to design individual primer pairs that would detect one specific transcript variant alone). The genes *ZNF189* and *BCHE* satisfied all criteria and their transcript variants could be successfully amplified, serving as a proof-of-principle target (Fig 2A). The qPCR

**Table 2. Enriched GO pathway clusters.**

| | Pathway | p-value |
|---|---|---|
| GO biological process | | |
| | regulation of cell development | $6.62 \cdot 10^{-14}$ |
| | regulation of nitric oxide biosynthetic process | $1.96 \cdot 10^{-10}$ |
| | mitotic cell cycle | $6.43 \cdot 10^{-10}$ |
| | regulation of transport | $1.71 \cdot 10^{-08}$ |
| | nuclear DNA replication | $1.20 \cdot 10^{-06}$ |
| | regulation of cellular component size | $1.46 \cdot 10^{-06}$ |
| | phosphate-containing compound metabolic process | $5.51 \cdot 10^{-06}$ |
| | single-organism catabolic process | $5.65 \cdot 10^{-06}$ |
| | negative regulation of transcription, DNA-templated | $2.45 \cdot 10^{-05}$ |
| | neurotrophin TRK receptor signaling pathway | $3.52 \cdot 10^{-05}$ |
| GO molecular functions | | |
| | identical protein binding | $7.26 \cdot 10^{-16}$ |
| | nucleic acid binding transcription factor activity | $2.15 \cdot 10^{-10}$ |
| | ubiquitin-protein transferase activity | $1.78 \cdot 10^{-08}$ |
| | protein kinase binding | $7.60 \cdot 10^{-08}$ |
| | zinc ion binding | $6.15 \cdot 10^{-06}$ |
| | substrate-specific transporter activity | $7.63 \cdot 10^{-05}$ |
| | transcription cofactor activity | $8.21 \cdot 10^{-05}$ |
| | protein serine/threonine kinase activity | $1.26 \cdot 10^{-03}$ |
| | DNA-directed DNA polymerase activity3 | $1.82 \cdot 10^{-03}$ |
| | Ras guanyl-nucleotide exchange factor activity | $2.41 \cdot 10^{-03}$ |
| GO cellular component | | |
| | perinuclear region of cytoplasm | $2.32 \cdot 10^{-04}$ |
| | nuclear speck | $9.90 \cdot 10^{-04}$ |
| | nuclear chromosome part | $4.04 \cdot 10^{-03}$ |
| | plasma membrane part | $1.33 \cdot 10^{-02}$ |
| | intercellular bridge | $1.56 \cdot 10^{-02}$ |
| | cell projection | $1.74 \cdot 10^{-02}$ |
| | nuclear envelope | $1.95 \cdot 10^{-02}$ |
| | nucleolus | $3.10 \cdot 10^{-02}$ |
| | membrane protein complex | $3.18 \cdot 10^{-02}$ |

Displayed are the titles of each pathway cluster. A cluster consists of multiple pathways that share a set of genes and have shown high overlap. Only significant pathways after correction have been considered for the clustering. The list of clusters is sorted by the aggregated p-values of each pathway in one cluster.

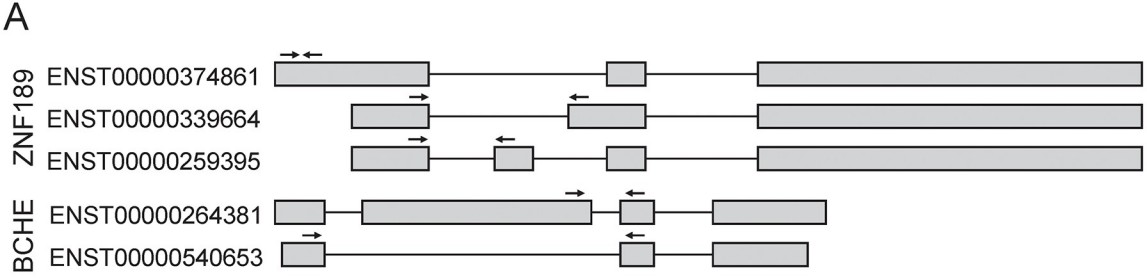

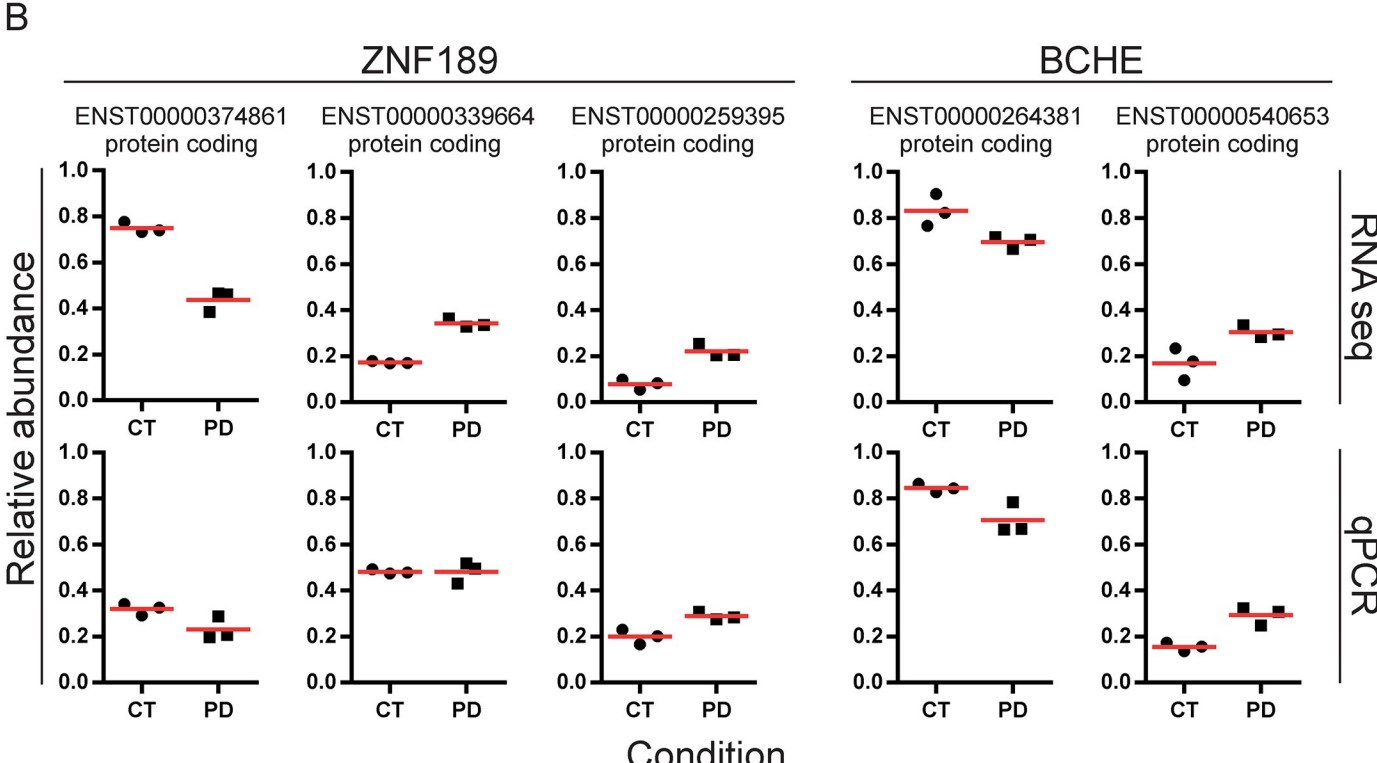

**Fig 2. qPCR validation of *ZNF189* and *BCHE* relative transcript abundances in individuals with PD and controls.** A: Schematic representation of ZNF189 and BCHE transcript variants analysed by qPCR. qPCR primer positions are indicated by arrows. B: Comparison of relative transcript abundances for the genes *ZNF189* and *BCHE*, obtained from RNASeq and qPCR. The upper row represents raw relative transcript abundances. Listed are only transcripts that remained after filtering. Data points are grouped by condition on the x-axis (PD vs CT). The three data points per group represent the three samples selected for qPCR. The lower row represents the results of qPCR analysis. Red lines show the mean of the respective group.

analysis replicated the results of the RNA-Seq-based DTU analyses for two of the three isoforms of *ZNF189* (ENST00000374861 and ENST00000259395), while the third isoform (ENST00000339664) appeared unchanged (Fig 2B). The qPCR analysis for *BCHE* confirmed the increased relative expression of isoform ENST00000540653 and the decreased relative expression of isform ENST00000264381 (Fig 2B).

## Pre-filtering reduces transcriptome complexity

To reduce the false discovery rate (FDR), transcripts and genes underwent a pre-filtering based on a minimum expression level prior to the analysis (see Methods). This pre-filtering

affected the distribution of mean transcript expression and the mean number of transcripts per gene. In the discovery cohort, 77% ($n = 137, 437$) of all transcripts and 75% ($n = 38, 100$) of all genes were removed due to insufficient expression. Likewise, 82% ($n = 143, 823$) of all transcripts and 78% ($n = 39, 342$) of all genes were filtered out in the replication cohort. The distribution of mean transcript expression in the discovery cohort was shifted from a median of 15 read counts to 61, and from 12 to 63 in the replication cohort, after excluding low expressed transcripts and genes. The filtering procedure reduced the standard deviation of the mean transcript distribution in both cohorts from 30,753 to 432 in the discovery cohort, and from 39,096 to 484 in the replication cohort (Fig 3A). We also observed a reduction in the median number of transcripts per gene, from 9 to 3 in the discovery cohort and from 10 to 3 in the replication cohort (Fig 3B). We also observed an increase in the relative amount of protein coding transcripts as well as a decrease in the amount of pseudogene transcripts, snoRNAs, snRNAs, miRNAs and rRNAs (Fig 3C).

## Alternative DTU methods agree in effect size and are minimally influenced by accounting for cell type composition

We investigated the agreement of effect size (i.e., the modeled coefficient for the disease state) in terms of magnitude and direction between the two tools in the discovery cohort. Overall, both methods agreed on the estimated effect size ($R = 0.97$, $p = 2.2 \cdot 10^{-16}$, $n = 40, 520$) and the concordance was even more pronounced in the subset of DTU events that were significant for either one of the cohorts ($R = 0.98$, $p = 2.2 \cdot 10^{-16}$, $n = 813$) (Fig 4). The general trend of statistical significance showed that transcripts which were identified as DTU events by at least one of the methods were likely to be defined at least as nominally significant by the alternative method: 97% of all DRIMSeq DTU events were nominally significant according to DEXSeq and 98% of all DEXSeq DTU events were reported as nominally significant by DRIMSeq. The concordance between the two methods in the replication cohort is shown in S2 Fig. We have recently shown that cell type heterogeneity can have a substantial impact on DGE analyses in bulk brain tissue [18]. To determine whether this also applied to our DTU analyses, we assessed the effect of accounting for cell type composition on our results. To this end, we obtained relative cellularity estimates (marker gene profiles, MGPs) for the cortical cell-types that were shown to be significantly associated with disease status (oligodendrocytes and microglia) in our previous study employing the same samples [18]. Accounting for cellular composition slightly increased the discovery signal, identifying a few more DTU genes with both DRIMSeq and DEXSeq. This effect was minor, however, as most DTU genes and events were identified irrespective of whether cell-type composition was accounted for or not (S3 and S4 Figs).

## Most DTU events are not detected by conventional DGE analysis

Next, we sought to determine whether DTU events were detectable at the gene level by comparing the results of the DTU analysis to a conventional DGE analysis performed on the same dataset [18]. We found that less than 3% ($n = 13$) of the DTU genes ($n = 584$) were also significant at the gene level (BH corrected, $FDR < 0.05$) (Fig 1A), suggesting that compensatory changes across transcripts can balance out overall gene expression. Indeed, in genes with two DTU events, the effect size of these generally tended to move in opposite directions, canceling out the change in overall gene expression (Fig 5A). Similarly, in genes with only one DTU event, the effect size of DGE was smaller than the effect size of DTU, or even close to zero (Fig 5B), which likely originated from compensation distributed across multiple transcripts.

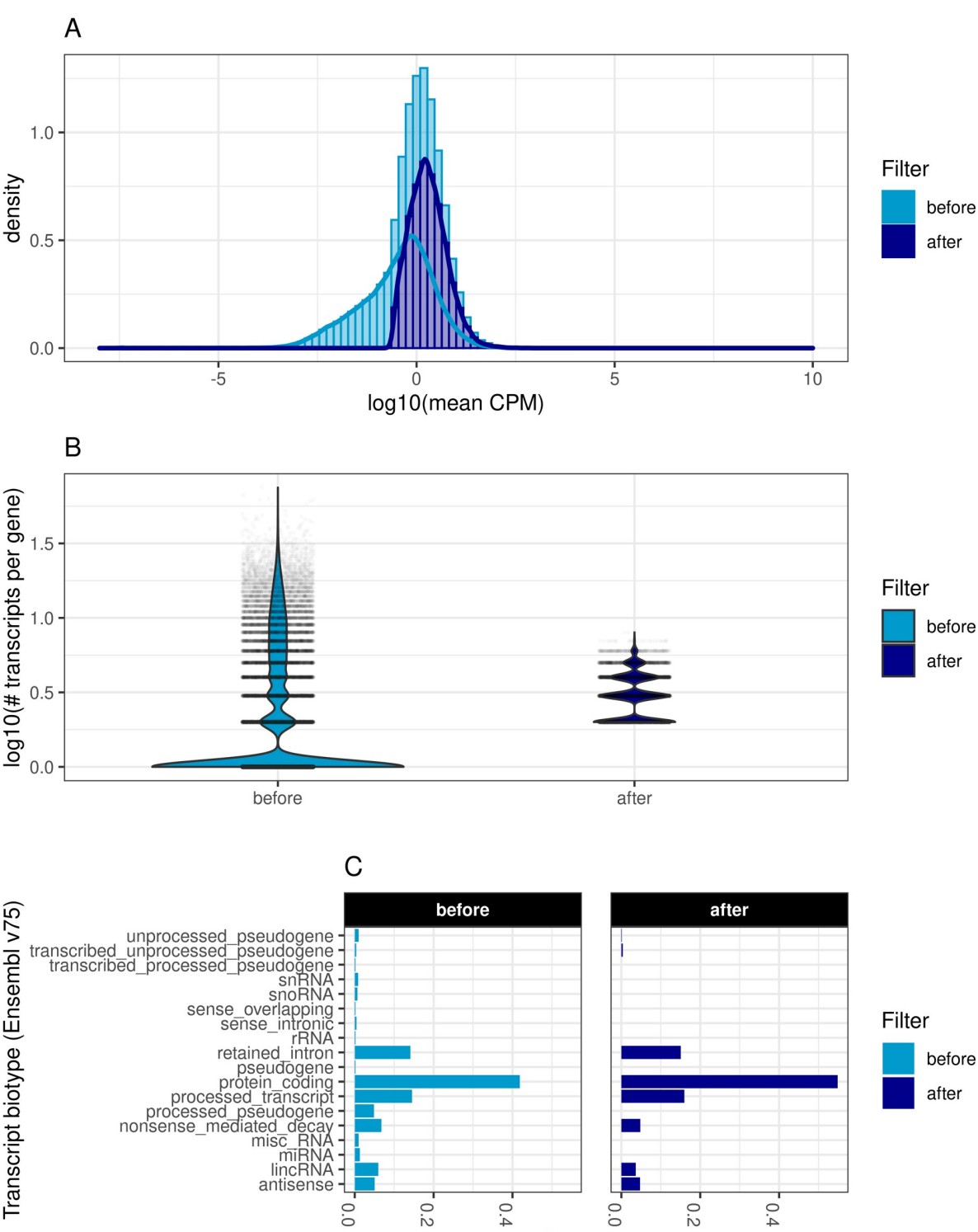

**Fig 3. Transcript filter statistics.** Comparison of data distributions before and after filtering of low expressed transcripts and genes. A: Log10 of the mean CPMs (counts per million) over all samples before and after filtering out low-expressed transcripts and genes. B: Violin plots showing the distribution of the number of transcripts per gene (in logarithmic scale). Violin width is scaled by the total number of observations while jittered points represent actual observations. C: Bar plot of transcript biotypes as defined by Ensemble v75 before and after filtering. Displayed are the relative frequencies of each category normalized by the number of transcripts before and after filtering. Categories with frequencies smaller than 0.001 were excluded for better visualization.

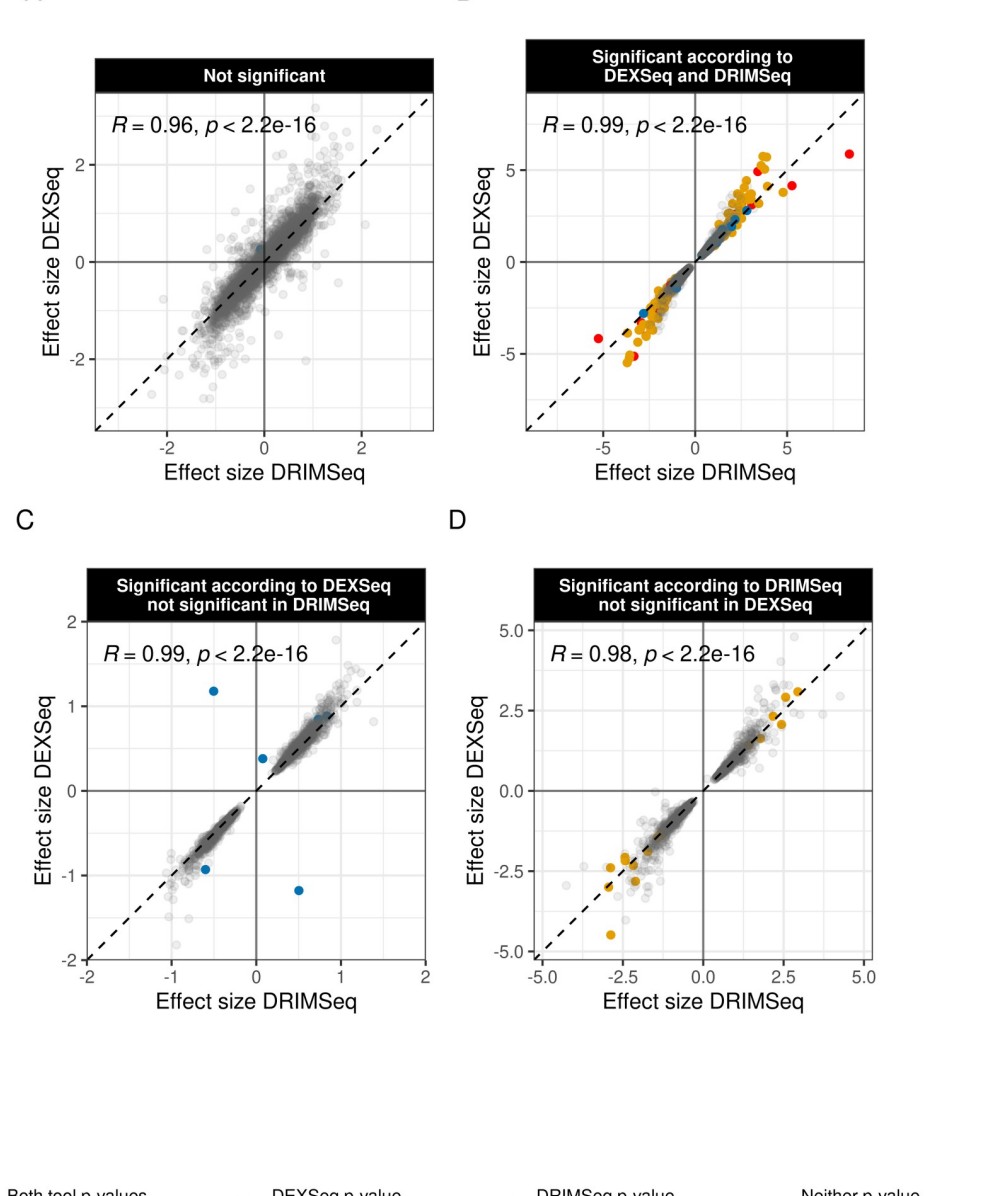

**Fig 4. Concordance between DEXSeq and DRIMSeq.** Estimated transcript usage effect sizes are shown for each transcript of the discovery cohort, with results from each tool on each of the axes (DRIMSeq x-axis, DEXSeq y-axis). Points situated on the diagonal represent transcripts with equal effect size in both tools. Points situated inside the first and third quadrant represent transcripts agreeing in direction across tools (i.e. first quadrant: up-regulated in PD, third quadrant: down-regulated in PD). A: Transcripts that did not reach statistical significance in the DTU analyses by either DRIMSeq or DEXSeq. B: Transcripts found to be significant by both tools. C: Transcripts found to be significant by DEXSeq only. D: Transcripts found to be significant by DRIMSeq only. Transcripts identified as DTU events (significant after p-value adjustment) are coloured according to the plot legend. Red: DTU event by both tools, blue: DTU event by DEXSeq only, yellow: DTU event by DRIMSeq only, grey: transcript either did not survive FWER correction in any of the tools or was not nominally significant.

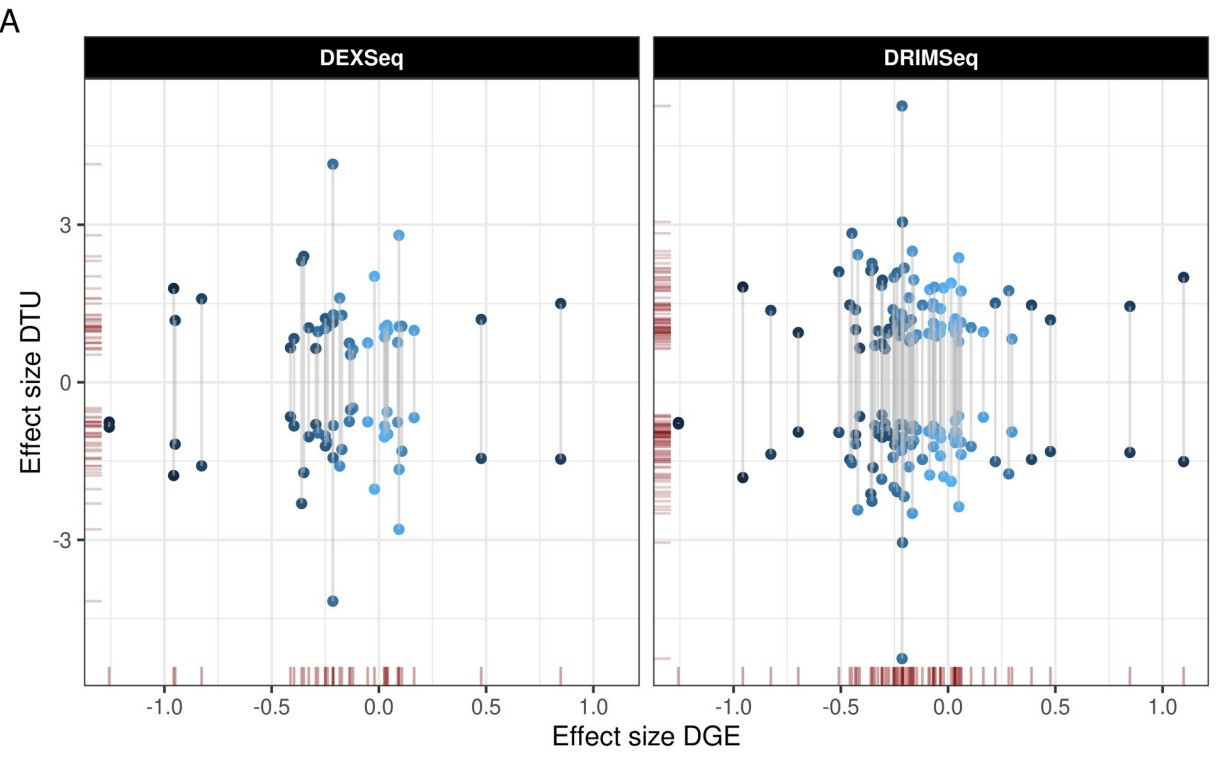

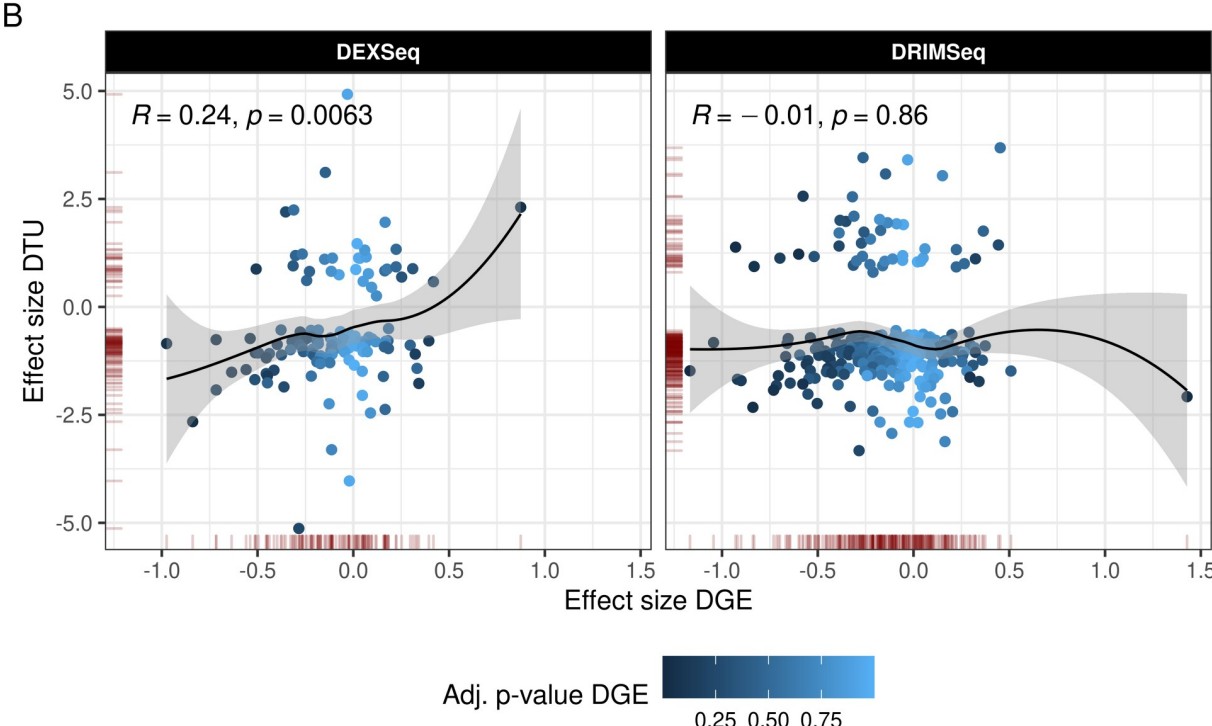

**Fig 5. Concordance with DGE.** The plot shows the relationship between the DTU effect size for each transcript (y-axis) and DGE effect size (x-axis). Data points correspond to transcripts. The x-coordinate of each point represents the effect size estimated for its parent gene in DGE analysis. The color scale indicates DGE significance after correction. A: DTU genes with 2 DTU events, with connected points representing each of the events from the gene. B: DTU genes with a single DTU event.

**Table 3. DTU genes detected by DGE.**

| Tool | Gene | Transcript ID | Biotype | ES DTU | ES DGE |
|---|---|---|---|---|---|
| DRIMSeq | BCHE | ENST00000540653 | protein coding | 1.81 | -0.96 |
| DRIMSeq | BCHE | ENST00000264381 | protein coding | -1.81 | -0.96 |
| DEXSeq | DAAM2 | ENST00000491083 | processed transcript | 0.99 | -0.80 |
| DRIMSeq | EAF1-AS1 | ENST00000610011 | antisense | 2.79 | 0.99 |
| DRIMSeq | FRG1B | ENST00000439954 | protein coding | -0.77 | -1.26 |
| DRIMSeq | FRG1B | ENST00000358464 | protein coding | -0.79 | -1.26 |
| DRIMSeq | FRG1B | ENST00000479318 | nonsense mediated decay | 1.13 | -1.26 |
| DRIMSeq | FRMPD2 | ENST00000491130 | retained intron | -2.97 | -1.17 |
| DRIMSeq | FRMPD2 | ENST00000486151 | retained intron | 2.97 | -1.17 |
| DRIMSeq | HIBCH | ENST00000414928 | nonsense mediated decay | 1.58 | 0.88 |
| DRIMSeq | MIA | ENST00000597600 | protein coding | -2.08 | 1.43 |
| DRIMSeq | MIA | ENST00000593317 | retained intron | 2.08 | 1.43 |
| DRIMSeq | MMP24-AS1 | ENST00000566203 | antisense | 0.91 | -0.58 |
| DRIMSeq | PRODH | ENST00000334029 | protein coding | -1.48 | -1.17 |
| DRIMSeq | SLCO1A2 | ENST0000452078 | protein coding | -0.83 | -1.04 |
| DRIMSeq | SLCO1A2 | ENST00000463718 | retained intron | 0.83 | -1.04 |
| DRIMSeq | TSPAN15 | ENST00000475069 | retained intron | 2.32 | -0.84 |
| DRIMSeq | TSPAN15 | ENST00000373290 | protein coding | -2.32 | -0.84 |
| DRIMSeq | UFSP2 | ENST00000509180 | protein coding | 1.22 | -0.60 |
| DRIMSeq | VWF | ENST00000538635 | processed transcript | -1.38 | -0.93 |
| DRIMSeq | VWF | ENST00000261405 | protein coding | 1.38 | -0.93 |

Each transcript is described by its Ensemble identifier (version 75). The effect size (ES) is relative to the controls, i.e. positive ES represents an increase in PD relative to controls, negative ES a decrease. All entries in the table represent DTU events of which the parent gene was detected by DGE (BH adjusted, FDR = 0.05). DTU events that were identified by both DRIMSeq and DEXSeq are listed only with the estimated ES of DRIMSeq. The list is sorted by gene name in alphabetical order.

Only 13 DTU genes with at least one DTU event were also identified by DGE (Table 3). Six of these genes had a single DTU event and the remaining 7 had multiple DTU events. Of the 6 genes with a single DTU event, 3 showed the same direction of change in both DGE and DTU, whereas in the other 3, DGE and DTU indicated changes in opposite directions. For all 7 DTU genes with multiple DTU events, at least one DTU event was in the opposite direction of the DGE change. For example, while the protein coding transcript of the *VWF* gene was up-regulated, DGE analysis showed down-regulation at the gene-level, driven by a non-protein coding isoform. These results indicate that DTU analyses provide important additional insight into the transcriptomic landscape of PD.

## Detected DTU events replicated in an independent patient cohort

We replicated our findings using RNA-Seq data from an independent cohort from the Netherlands Brain Bank (*n* = 10/11 PD/controls; Table A in S1 File). A total of 32,040 transcripts passed quality filtering in the replication cohort. The majority of these (*n* = 29, 807; 93%) overlapped with the pre-filtered transcripts of the discovery cohort and were further analyzed for replication. A total of 10,713 transcripts from the discovery cohort, however, did not pass prefiltering in the replication cohort. Of these, 249 were identified as DTU events in the discovery cohort (S5A Fig). To assess the overall concordance between the two cohorts, we divided the common set of transcripts into 4 categories according to their nominal significance in differential usage in PD: i. non-significant in either cohort, ii. significant only in the discovery cohort, iii. significant in both cohorts, iv. significant only in the replication cohort. For each

category we assessed the concordance in DTU direction between the discovery and replication cohort (Fig 6A). In the group of non-significant transcripts, we observed a low correlation in the direction of DTU (Pearson's $R = 0.07$, $p = 2.2 \cdot 10^{-16}$, $n = 2,5002$), with only 54% of transcripts agreeing between the cohorts. A higher correlation (Pearson's $R = 0.19$, $p = 2.2 \cdot 10^{-16}$, $n = 3776$) was observed for the group of transcripts which were nominally significant in the discovery cohort only, where 59% of transcripts showed the same direction of change in both cohorts. Transcripts which were significant only in the replication cohort showed no correlation (Pearson's $R = 0.058$, $p = 0.092$, $n = 843$) in the direction of DTU. The highest correlation (Pearson's $R = 0.25$, $p = 0.6 \cdot 10^{-3}$, $n = 186$) was observed in the group of transcripts that were nominally significant in both cohorts, with a 62% concordance in direction.

When we reduced the collection of transcripts to DTU events detected in the discovery cohort, we saw a high correlation (Pearson's $R = 0.28$, $n = 481$, $p = 2.5 \cdot 10^{-10}$), with 64% of these transcripts agreeing on the direction of change. This suggests that highly significant DTU events identified in our discovery cohort show a similar trend in our replication cohort (Fig 6B). Notably, 23% of the DTU genes identified in the discovery cohort were filtered out during pre-processing of the replication cohort and thus were excluded from this analysis.

A total of 23 DTU events in 19 genes detected in the discovery cohort were concordant in direction of change and nominally significant in the replication cohort (Table 4).

Among the 19 replicated DTU genes, 15 showed one DTU event and four comprised two DTU events per gene. Interestingly, in the four genes exhibiting two DTU events (*LINC00499*, *BCHE*, *THEM5*, *SLC16A1*), these moved in opposite directions. In *BCHE* and *THEM5*, DTU resulted in isoform switches (i.e. two DTU events in opposite directions) between different protein-coding transcripts. *THEM5*, encoding an acyl-CoA thioesterase involved in mitochondrial fatty acid metabolism, showed decreased usage of the full-length transcript (encoding a 247 amino acid protein) and increased usage of a shorter transcript (encoding a 119 amino acid protein) in PD. The down-regulated, full-length isoform was predicted to localize to the mitochondria (*likelihood* = 0.99), whereas the up-regulated, shorter isoform was more likely to localize to the extracellular space (*likelihood* = 0.36) than to the mitochondria (*likelihood* = 0.21). Hence, the decreased usage of the full-length isoform could result in a decrease of mitochondrial *THEM5* activity in PD. A similar pattern was observed for the *BCHE* gene, encoding a butyrylcholinesterase, with the full-length isoform (encoding a protein of 602 amino acids) down-regulated in PD, and an up-regulated shorter transcript encoding a putative protein of 64 amino acids. While both isoforms were predicted to be soluble and localize to the extracellular space, the shorter isoform lacks the substrate binding site located at positions 144 and 145 and it is therefore predicted to be non-functional, suggesting that *BCHE* function may be down-regulated in PD. The *SLC16A1* gene, encoding a lactate transporter in oligodendroglia, showed a switch from a protein-coding to a non-protein coding isoform in PD, revealing decreased expression of the protein coding transcript in PD.

In agreement with the down-regulation observed at the gene level, only 2 out of 19 replicated genes with DTU showed a significant altered overall gene expression: *BCHE* and *PRODH* (BH corrected, *FDR* < 0.05). In the case of *BCHE*, the down-regulation was observed for the full-length transcript as described above. *PRODH* exhibited a single DTU event consisting of a decreased relative expression of a protein-coding transcript variant in PD.

## No evidence of DTU for genes linked to monogenic PD

Previous research had suggested that genes linked to monogenic PD, including *SNCA*, *PARK7* and *PRKN*, may exhibit altered transcript expression patterns in idiopathic PD [11, 12, 14]. Therefore, we sought to investigate whether these observations replicate in our data.

A

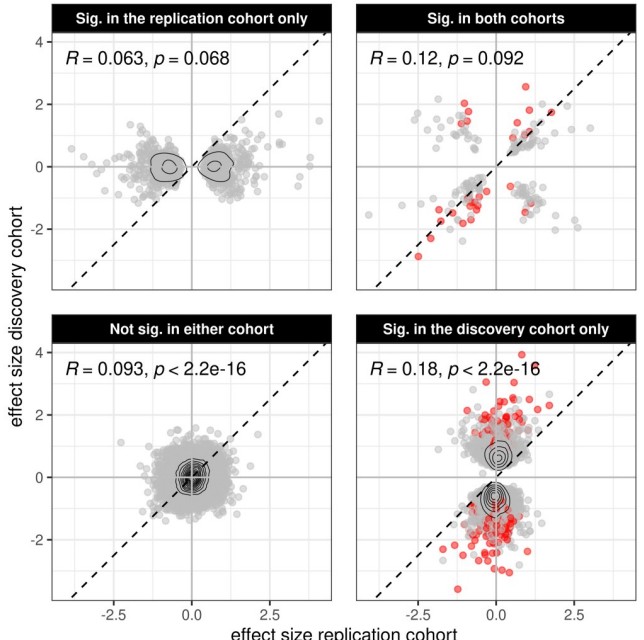

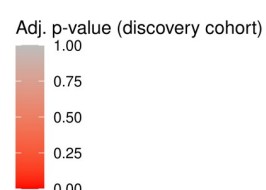

B

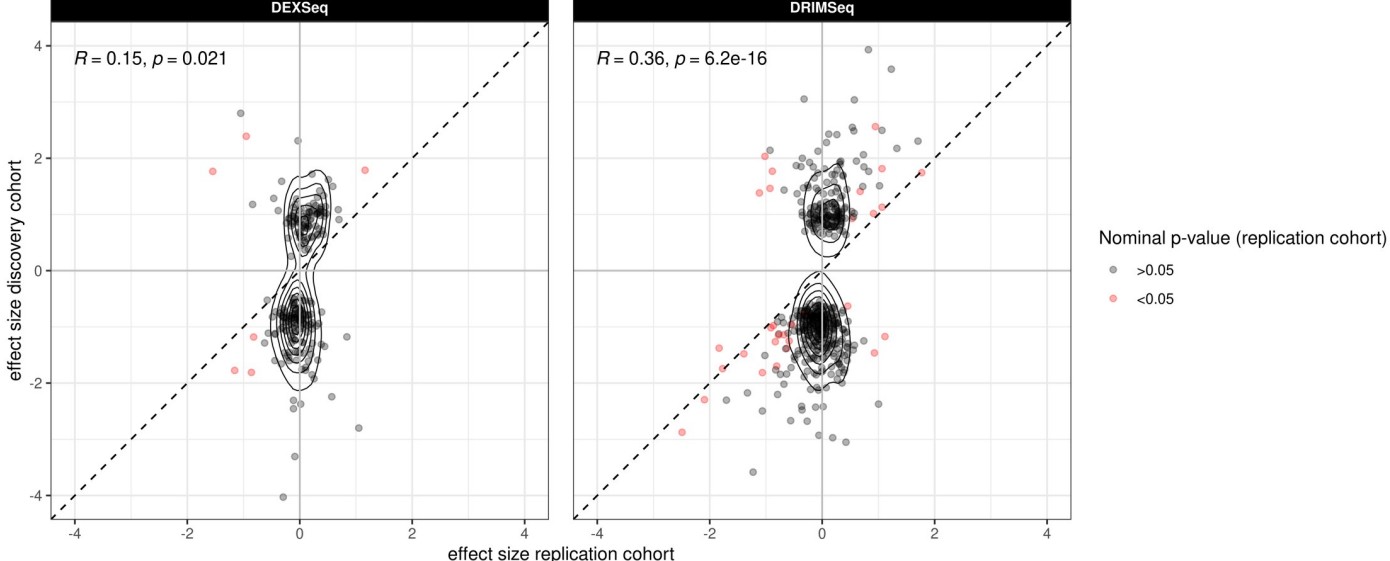

**Fig 6. DTU replication in an independent cohort.** Each data point corresponds to one transcript. The estimated effect size in the discovery cohort is represented on the y-axis and the estimated effect size for the replication cohort on the x-axis. A: The overlapping set of transcripts between the two cohorts is divided into 4 categories depending on their nominal significance in either cohort according to DRIMSeq. Transcripts not found to be significant in any cohort are shown in the lower left quadrant. Transcripts found to be significant only in the replication or in the discovery cohort are in the upper left and lower right quadrant respectively. The color scale (red-gray) shows adjusted p-value in the discovery cohort. B: Displayed are all DTU events (significant after correction) of the discovery cohort. Red color indicates nominal significance (before correction) in the replication cohort. The two columns present the results obtained from each respective tool.

**Table 4. Replicated DTU genes.**

| Tool | Gene | Transcript ID | Biotype | ES discovery cohort | ES replication cohort |
|------|------|---------------|---------|---------------------|------------------------|
| DEXSeq | BCHE | ENST00000264381 | protein coding | -1.78 | -1.16 |
| DEXSeq | BCHE | ENST00000540653 | protein coding | 1.79 | 1.16 |
| DEXSeq | XPA | ENST00000375128 | protein coding | -1.18 | -0.82 |
| DEXSeq | VWA9 | ENST00000573314 | nonsense mediated decay | -1.81 | -0.86 |
| DRIMSeq | SLC16A1 | ENST00000369626 | protein coding | -1.02 | -0.91 |
| DRIMSeq | SLC16A1 | ENST00000478835 | processed transcript | 1.02 | 0.91 |
| DRIMSeq | THEM5 | ENST00000453881 | protein coding | 1.74 | 1.77 |
| DRIMSeq | THEM5 | ENST00000368817 | protein coding | -1.74 | -1.77 |
| DRIMSeq | BCHE | ENST00000540653 | protein coding | 1.81 | 1.06 |
| DRIMSeq | BCHE | ENST00000264381 | protein coding | -1.81 | -1.06 |
| DRIMSeq | HDAC3 | ENST00000305264 | protein coding | -1.38 | -0.63 |
| DRIMSeq | XPA | ENST00000375128 | protein coding | -1.15 | -0.78 |
| DRIMSeq | ZNF208 | ENST00000601993 | protein coding | 1.13 | 1.07 |
| DRIMSeq | VWA9 | ENST00000573314 | nonsense mediated decay | -1.70 | -0.81 |
| DRIMSeq | SLC2A4RG | ENST00000473157 | processed transcript | -2.30 | -2.09 |
| DRIMSeq | CD46 | ENST00000367041 | protein coding | -1.25 | -0.59 |
| DRIMSeq | ST3GAL5 | ENST00000393808 | protein coding | -0.96 | -0.54 |
| DRIMSeq | ACO1 | ENST00000379923 | protein coding | -1.38 | -1.83 |
| DRIMSeq | PRODH | ENST00000334029 | protein coding | -1.48 | -1.39 |
| DRIMSeq | RPS9 | ENST00000391752 | protein coding | 2.56 | 0.95 |
| DRIMSeq | LRTOMT | ENST00000440313 | protein coding | 1.41 | 0.68 |
| DRIMSeq | LINC00499 | ENST00000510736 | lincRNA | -1.14 | -0.68 |
| DRIMSeq | RNF38 | ENST00000377885 | protein coding | -0.98 | -0.87 |
| DRIMSeq | APIP | ENST00000527830 | processed transcript | -2.87 | -2.49 |
| DRIMSeq | CNPY2 | ENST00000548013 | retained intron | -0.79 | -0.31 |
| DRIMSeq | LINC00499 | ENST00000502757 | lincRNA | 0.93 | 0.54 |
| DRIMSeq | ACAA1 | ENST00000452171 | protein coding | -1.26 | -0.83 |

Each transcript is described by its Ensemble identifier (version 75). The effect size (ES) is relative to the controls, i.e. positive ES represents an increase in transcript usage in PD relative to controls, negative ES a decrease. The p-value as reported by stageR for each tool separately (DEXSeq and DRIMSeq) is representative for the level of significance after FWER control with $\alpha = 0.05$ and is lower than 0.03 for all listed DTU events. The table is sorted by the p-value in increasing order and grouped by the tool that identified the transcript

Increased expression of four *SNCA* transcript variants, encoding the protein isoforms SNCA-140, SNCA-126, SNCA-112 and SNCA-98, were reported in the prefrontal cortex of individuals with PD [12]. None of these transcripts showed evidence of DTU in our analysis. The transcript (ENST00000506244) encoding the full-length protein (SNCA-140), showed a trend for reduced relative expression in PD, but this did not reach statistical significance ($p = 0.055$, effect size = −0.48, DRIMSeq). In the same study, two out of seven protein-coding splice variants of PRKN (TV3 and TV12) were suggested to be overexpressed in the PD brain. In our data, only two PRKN transcript variants (TV1 and TV2) showed sufficient expression to be analyzed, and neither of them showed statistical evidence of DTU (nominal $p > 0.79$, absolute effect size $< 0.09$, DRIMSeq) in agreement with the results reported in [12].

Finally, one study reported that the altered relative transcript abundance of *PARK7* in blood may be used as a biomarker for PD [14]. None of the transcript variants of *PARK7* were sufficiently expressed in our dataset to investigate the transcript usage pattern of this gene in the PD brain.

## Discussion

We report the first transcriptome-wide DTU study in PD. Our analyses reveal that multiple DTU events occur in the PD brain and many of these are predicted to have a functional impact. Interestingly, the vast majority of genes exhibiting DTU are not detected by conventional DGE analysis on the same dataset. This is either because DTU occurs in low-expressed isoforms, or due to antagonistic, inverse changes in other transcripts of the same gene, canceling out the net change at the gene expression level.

Our findings suggest that DTU events in PD may have important downstream consequences for protein function, irrespective of whether there is a measurable difference in the total gene expression levels. Changes in the relative expression of different transcripts of a gene affect the ratio of the resulting protein isoforms and could, therefore, influence biological processes through variation in function and/or subcellular localization. Moreover, switches may occur between protein coding and non-coding transcript isoforms, thereby affecting the overall protein level. Changes in the usage ratios of low expressed and/or non-protein coding isoforms may also have important biological effects, as it has been shown that these are highly cell- and tissue-specific, and have a substantial impact on the composition and function of the proteome [5].

In our dataset, individuals with PD showed a significant decrease in the relative usage of a *THEM5* transcript variant that encodes the full-length THEM5 protein isoform, predicted to localize to mitochondria. This isoform is involved in mitochondrial fatty acid metabolism by exhibiting esterase activity with a preference for long and unsaturated fatty acid-CoA esters [19]. Decreased *THEM5* function has been shown to influence the remodeling process of mitochondrial inner membrane cardiolipin [19, 20], resulting in abnormal mitochondrial morphology and impaired mitochondrial respiration [19], both of which occur in PD [18, 21]. A concomitant increase in the relative expression of a shorter THEM5 isoform resulted in relatively unchanged levels of total gene expression. However, as this isoform encodes a protein lacking the first 37 N-terminal amino acids, it is unlikely to localize to mitochondria, and may therefore not replace the full-length protein functionally [19].

A protein-coding transcript of the *SLC16A1* gene was significantly down-regulated in the PD brain and accompanied by an increase of similar magnitude in a non-protein coding transcript. *SLC16A1* encodes a monocarboxylate transporter (*MCT1*) responsible for lactate and pyruvate trafficking across cell membranes. *MCT1* is the most abundant lactate transporter in the central nervous system, where it is highly expressed in oligodendroglia. It has been shown that *MCT1* plays a key role in the energy homeostasis of neurons, by regulating lactate transport between oligodendroglia and axons. *MCT1* disruption causes axonal dysfunction and neurodegeneration in cell and animal models and MCT1 levels have been found to be decreased in patients and mouse models of ALS [22, 23].

Another gene of interest was *BCHE*, which showed a decreased usage of the protein-coding full-length transcript, suggesting that the level of the functional full length protein isoform may be decreased in PD. Interestingly, genetic variation in this gene has been associated with Alzheimer's disease [24], susceptibility to pesticide toxicity [25] and, more recently, with PD [26].

In the few genes that were detected by both DTU and DGE analysis, DTU provided additional functional insight. Since changes in the relative isoform expression can occur in opposite directions to the overall gene-level expression, transcript-level resolution is essential in order to predict the functional consequences of altered expression.

Our analyses did not confirm a previous report of altered transcript expression in the *SNCA* gene in the PD frontal cortex [12]. These findings were based on a small PD cohort

(n = 5) with no reported neuropathological confirmation of the diagnosis. The fact that the reported transcripts were confidently detected in our data but showed no evidence (or trend) of altered relative expression in either of our cohorts, suggests that this effect, if real, is not a general or common phenomenon in PD. Alternatively, the lack of replication may reflect different genetic backgrounds and environmental exposures in different populations (Spanish, Norwegian and Dutch). The *PRKN* transcripts TV3 and TV12, which were reported to show altered expression in PD in the same sample as *SNCA* [12, 13] did not show sufficient expression in our material to be confidently assessed for replication.

While most identified DTU genes in our results do not have a known role in PD, pathway analyses showed significant enrichment in clusters associated with the pathophysiology of PD, including reactive oxygen species (ROS) generation and protein degradation. These results confirm that our findings are related to the biology of PD and highlight DTU analyses as a complementary strategy to nominating novel disease candidate genes and processes.

A potential limitation in our study is posed by differences in cell-type composition between brain tissue of patients and controls. We have recently shown that this can be an important confounding factor in differential expression analysis of bulk brain tissue [18]. To mitigate this problem, we accounted for differences in cellularity across samples by including cell type estimates for specific cell types found to be significantly associated with disease status, as covariates in our model. Notably, correcting for cell-type composition had only a minor effect in our results, supporting the notion that most identified DTU events are not driven by differences in cellularity between PD and controls.

While our top DTU findings replicate across the two independent cohorts, suggesting these changes are robustly associated with PD, we nevertheless observe an overall low concordance between the cohorts. This most likely reflects a combination of biological and technical factors, including limited power due to the relatively small sizes of the cohorts, heterogeneous disease biology and cell-composition, population-specific and/or brain bank-specific effects, differences in the age and RIN ranges. Differences between the cohorts were also evident in the filtering results, whereby a larger number of transcripts in the replication cohort were filtered out in comparison to the discovery cohort, as summarized in S5A Fig. We hypothesized that this may be related to the overall higher RINs of the samples from the replication cohort. Transcripts which were detected in the discovery cohort but not in the replication cohort showed a negative correlation with RIN (S5B Fig), suggesting that lower RNA quality (reflected by lower RIN values) is associated with higher transcript counts due to an increase in non-specific alignments in degraded samples.

Further replication in larger samples will be required in order to confirm and further dissect the DTU landscape of the PD brain. Methodological limitations should also be considered. While DRIMSeq was designed specifically for DTU analysis and assesses the relationship of each transcript abundance relative to the total transcriptional output, it may have difficulties to correctly estimate the dispersion for genes with a large number of isoforms [16]. This can potentially lead to inaccurate transcript proportion estimations and increase the susceptibility to false positive results, as suggested by the p-value distributions. Conversely, DEXSeq cannot capture the transcript-gene relationship directly, which might explain its general lower sensitivity compared to DRIMSeq.

## Conclusion

In conclusion, our findings provide the first insight into the DTU landscape of PD. We show that DTU is a prominent feature in the PD brain and may have important functional consequences by altering the structural and functional composition of the proteome. We therefore propose that

DTU analyses should be an essential component of transcriptomic studies, along with DGE analyses, because they provide additional insight into the transcriptomic landscape and allow a more accurate prediction of the functional consequences of detected changes in gene expression.

## Methods

### Cohorts

Fresh-frozen prefrontal cortex tissue (Brodmann area 9) was available from two independent cohorts. The discovery cohort comprised individuals with idiopathic PD ($n = 17$) from the Park West study, a prospective population-based cohort, which has been described in detail [15], and demographically matched controls ($n = 11$). Samples were collected and stored in our Brain Bank for Aging and Neurodegeneration. The replication cohort comprised individuals with idiopathic PD ($n = 10$) and demographically matched controls ($n = 11$) from the Netherlands Brain Bank. The details of the cohorts are summarized in Table A in S1 File.

### Ethics statement

Ethical permission for these studies was obtained from our regional ethics committee "Regional Committee for Medical and Health Research Ethics": REK 2017/2082, 2010/1700, 131.04 (REC, https://rekportalen.no/). Written formal informed consent was obtained from all participants or their next of kin.

### RNA sequencing

Total RNA was extracted from prefrontal cortex tissue homogenate for all samples using RNeasy plus mini kit (Qiagen) with on-column DNase treatment according to manufacturer's protocol. Final elution was made in 65 $\mu$l of dH2O. The concentration and integrity of the total RNA was estimated by Ribogreen assay (Thermo Fisher Scientific), and Fragment Analyzer (Advanced Analytical), respectively. Five hundred ng of total RNA was required for proceeding to downstream RNA-seq applications. First, ribosomal RNA (rRNA) was removed using Ribo-Zero™ Gold (Epidemiology) kit (Illumina, San Diego, CA) using manufacturer's recommended protocol. Immediately after the rRNA removal the RNA was fragmented and primed for the first strand synthesis using the NEBNext First Strand synthesis module (New England BioLabs Inc., Ipswich, MA). Directional second strand synthesis was performed using NEBNext Ultra Directional second strand synthesis kit. Following this the samples were taken into standard library preparation protocol using NEBNext DNA Library Prep Master Mix Set for Illumina with slight modifications. Briefly, end-repair was done followed by poly(A) addition and custom adapter ligation. Post-ligated materials were individually barcoded with unique in-house Genomic Services Lab (GSL) primers and amplified through 12 cycles of PCR. Library quantity was assessed by Picogreen Assay (Thermo Fisher Scientific), and the library quality was estimated by utilizing a DNA High Sense chip on a Caliper Gx (Perkin Elmer). Accurate quantification of the final libraries for sequencing applications was determined using the qPCR-based KAPA Biosystems Library Quantification kit (Kapa Biosystems, Inc.). Each library was diluted to a final concentration of 12.5 nM and pooled equimolar prior to clustering. 125 bp Paired-End (PE) sequencing was performed on an Illumina HiSeq2500 sequencer (Illumina, Inc.) at a target depth of 60 million reads per sample.

FASTQ files were trimmed using Trimmomatic [27] to remove potential Illumina adapters and low quality bases with the following parameters:

```
ILLUMINACLIP:truseq.fa:2:30:10
LEADING:3 TRAILING:3 SLIDINGWINDOW:4:15.
```

FASTQ files were assessed using fastQC [28] prior and following trimming.

## Transcript quantification

We used Salmon [29] with the fragment-level GC bias correction option (`--gcBias`) and the appropriate option for the library type (`-l ISR`) to quantify transcript expression in pseudo-alignment mode, using the GRCh37 genome as a reference. X and Y chromosomes were excluded from the GRCh37 reference genome, restricting quantification to transcripts located on autosomes.

Transcripts per million (TPM) values obtained with Salmon were scaled using the R package *tximport* [30] with the scaling method `scaledTPM`, the favored scaling method for DTU [31].

## DTU analyses and quality control

DTU analyses estimate transcript usage and detect changes in the relative contribution of a transcript to the overall expression of the gene. Transcript usage corresponds to the transcript-level expression counts of a transcript $i$ normalized by the sum of counts of all transcripts of a gene $j$:

$$TU_{i,j} = \frac{t_i}{\sum_{k=1}^{n_j} t_k} \ ,$$

(1)

where $n_j$ equals the number of transcripts of gene $j$ and $t_i$ is the expression count of transcript $i$. Hence, *differential* transcript usage describes a change in proportions between the groups (PD and controls).

For our analysis, we employed an alignment-free abundance estimation method [29], which enabled read quantification at the transcript level directly, as opposed to traditional read alignment methods that require bin or exon read counting and subsequent summarization to transcript level.

We performed DTU analysis between PD and controls using two alternative approaches implemented in the tools DRIMSeq [16] and DEXSeq [17]. While DEXSeq was designed for detecting differential exon usage, it is also suitable for assessing DTU by using estimated transcript abundances directly [6, 31, 32]. DRIMSeq was developed specifically for DTU analyses and is based on estimated transcript counts [16]. These methods assess alternative splicing by directly identifying transcripts that are differentially used, rather than detecting specific splice events. Both methods have shown comparable performance in benchmarks with simulated data [16, 31, 32]. A further advantage was that these tools allow for the inclusion of known covariates into the model design. DRIMSeq assumes a Dirichlet multinomial model for each gene and estimates a gene-wise precision parameter, whereas DEXSeq assumes a negative binomial distribution for counts of each transcript and estimates a transcript-wise dispersion parameter [31]. It is worth noting that DRIMSeq bases its analyses directly on the calculated transcript proportions, thereby modeling the correlation among transcripts in their parent-gene directly, whereas those correlations may not be accurately captured by DEXSeq, as it models each transcript separately and accounts for gene-transcript interaction with a covariate in its model design [31].

Due to the complexity of the human transcriptome in terms of diversity and number of transcripts per gene, DTU methodologies tend to exhibit a worse performance considering the false discovery rate (FDR) when compared to simpler organisms [6]. However, FDR can be reduced considerably if the collection of transcripts undergoes filtering prior to analysis [6]. Transcript filtering, in addition, alleviates the DRIMSeq-specific difficulty of capturing the full bandwidth

of transcript dispersion through the common gene-level dispersion estimate [16], which results otherwise in a decrease in performance for genes with increasing number of transcripts. We thus excluded lowly expressed transcripts with a soft filter, allowing for a certain percentage of all samples to have a transcript expression below the given threshold. This filtering methodology was chosen over hard filtering in order to avoid overlooking cases of DTU driven by lack of expression in one of the groups being compared, which would have been the case with a hard threshold filtering. Using the filtering method available in the DRIMSeq package, we excluded transcripts for which more than `n = min(#Controls, #PD)` samples did not reach 10 read counts or for which their relative contribution to the overall gene expression was smaller than one percent. In addition, we filtered out genes with less than 10 counts in any one sample. To investigate changes in transcript usage between PD and controls, the resulting filtered set of transcript-level counts were used as an input for both DEXSeq and DRIMSeq as recently suggested by [31]. Analyses were carried out independently on both cohorts.

## Model design

Sources of variation in our data were identified using principal component analysis (PCA) at the gene-level. RNA integrity number (RIN) correlated highly with the first principal component, indicating that RNA quality represents a major source of variation in the expression data.

Relative cellular composition in our samples was obtained from our previous study [18] using marker gene profiles (MGPs) [33, 34]. In summary, an MGP was calculated for each of the main cortical cell types (neurons, oligodendrocytes, astrocytes, endothelial, and microglia) by performing a PCA on the log-transformed expression (in counts per million) of cell type-specific marker genes from the NeuroExpresso database [33] and extracting the first principal component. MGPs for oligodendrocyte and microglia showed a significant association with the disease status (controls vs PD) and were accounted for in the DTU models together with RIN, gender, and age.

To explore the effect of accounting for disease-associated MGPs in the DTU results, we compared the two alternative designs, with and without oligodendrocyte and microglia MGPs. Accounting for cellular composition slightly increased the discovery signal, identifying a few more DTU genes with both DRIMSeq and DEXSeq. This effect was minor, however, as most DTU genes and events were identified irrespective of whether cell-type composition was accounted for or not (S3 and S4 Figs).

## Statistical testing

The results of the DTU analyses were further processed with StageR [35]. Gene-level aggregated p-values (q-values) as well as transcript-level p-values were passed to stageR for a two-stage screening of significance. For DEXSeq, nominal p-values of all transcripts of a gene were aggregated to a q-value and corrected using the function *perGeneQvalue*. For DRIMSeq, nominal p-values were already reported at the gene-level and further corrected within stageR using the Benjamini-Hochberg (BH) FDR procedure. To control the FWER, transcript-level significance was corrected within-gene, if the gene passed the first screening stage of stageR, with respect to the FDR controlled gene-level significance (q-value). Transcripts of genes which did not pass the first screening stage, were not further assessed for significance at the transcript-level. Nominal transcript-level p-values of both tools were adjusted within StageR using an adapted Holm-Shaffer family-wise error rate (FWER) correction method specifically designed for DTU analysis [35].

We define a transcript as a *DTU event*, if the FWER-controlled $p < \alpha$ with $\alpha = 0.05$. Similarly, we define as *DTU gene* any gene that exhibits at least one DTU event.

Similarly, we define $\alpha = 0.05$ for nominal significance.

## DTU pathway enrichment analysis

To assess the enrichment of DTU genes in predefined functional gene sets (pathways), we employed the *enrichment* function of the stringDB R package [36]. DTU genes identified in our discovery cohort were used as hits and all genes surviving the filtering step during pre-processing were used as background. Enrichment was tested for pathways defined by the Genome Ontology (GO) [37, 38]. Each of the three GO categories (Biological Process, Molecular Function, Cellular Compartment) was tested separately. To reduce redundancy of the top most enriched pathways ($FDR < 0.05$), we performed a clustering in each of the three GO categories. Pathways were clustered by iteratively joining nearest neighbors based on pathway similarity, which we defined with the Cohen's kappa coefficient ($\kappa$). The similarity of newly formed clusters and unvisited neighbours was iteratively recalculated, until no two clusters' $\kappa$ was higher than a chosen threshold of 0.4. Each cluster was given a representative title, chosen from the names of all the pathways in a cluster. The choice of the cluster title depended on the pathway size, pathway significance or chosen randomly if none of the previous criteria were sufficient. Finally, each pathway cluster was assigned a p-value by aggregating p-values of all cluster members with the Fisher method.

For specific cases of isoform switches between protein coding transcripts, we used the tool DeepLoc [39] to predict subcellular localization by retrieving the encoded amino acid sequence from the Ensembl release 75.

## RNA extraction, cDNA synthesis and quantitative PCR analysis

RNA extraction was carried out using the RNeasy Lipid Tissue Mini Kit (QIAGEN 74804), starting with ca. 20 mg brain tissue from three individuals with PD and three controls. 500 ng total RNA were subjected to cDNA synthesis using the SuperScript IV VILO Master Mix with ezDNase Enzyme (Thermofisher Scientific 11766500). Experiments were carried out in triplicates starting with a new cDNA synthesis from aliquoted total RNA. For the SYBR Green quantitative PCR analysis, the PowerUp SYBR Green Master Mix (Thermofisher Scientific, A25776) was used with a thermal cycling of one cycle at 95˚C for 20s and 40 cycles at 95˚C for 3s and 60˚C for 30s on a StepOnePlus instrument (Thermofisher Scientific), and with the primers listed in Table 5.

**Table 5. qPCR primer sequences.**

| Transcript ID | Primer name | Primer sequence |
|---|---|---|
| ENST00000374861 | ZNF189_374861 fw | 5'-TGGGGTTCGGGGTTGGGG-3' |
| ENST00000374861 | ZNF189_374861 rv | 5'-CGGTCACGACCCCAACAGC-3' |
| ENST00000339664 | ZNF189_339664 fw | 5'-GATGGCTTCCCCGAGCCC-3' |
| ENST00000339664 | ZNF189_339664 rv | 5'-ACACAGCCACATCCTCAAATG-3' |
| ENST00000259395 | ZNF189_259395 fw | 5'-GAGATGGCTTCCCCGAGCC-3' |
| ENST00000259395 | ZNF189_259395 rv | 5'-CTTATTTTCTCAGGCCGATTTATC-3' |
| ENST00000540653 | BCHE_540653 fw | 5'- GCAAACTTTGCCATCTTTGTTG-3' |
| ENST00000540653 | BCHE_540653 rv | 5'- CTTGTGCTATTGTTCTGAGTC-3' |
| ENST00000264381 | BCHE_264381 fw | 5'- AGATCCATAGTGAAACGGTGG-3' |
| ENST00000264381 | BCHE_264381 rv | 5'- CTTGTGCTATTGTTCTGAGTC-3' |
| | GAPDH | Assay ID Hs00266705_g1 (Thermofisher) |

## Supporting information

**S1 Fig. Diagnostic plots.** Data points in all plots represent one transcript, with coloring showing significant transcripts ($\alpha$ = 0.05) in red. P-values (uncorrected) are displayed as ($-log10$(p-value)). A: Volcano plot displaying the effect size (as estimated by the respective tool) in the x-axis and the p-value on the y-axis. Triangles mark extreme p-value outliers that were adjusted to fit into the plot. B: MA plot visualizing a transcript's significance as a function of its mean expression over all samples. C: Density ridges display the distribution of gene-level significance ($-log10$(p-value)) per gene type, where genes are grouped according to the number of transcripts they have after filtering. The color gradient was applied to visualize the p-value scale. The vertical dashed line corresponds to a p-value of 0.05.
(TIFF)

**S2 Fig. Concordance between DEXSeq and DRIMSeq in the replication cohort.** Estimated transcript usage effect sizes are shown for each transcript of the replication cohort, with results from each tool on each of the axes (DRIMSeq x-axis, DEXSeq y-axis). Points situated on the diagonal represent transcripts with equal effect size estimations of both tools; points situated inside the first and third quadrant of the coordinate system represent transcripts agreeing in direction according to both tools (i.e. up-regulated in PD: first quadrant, down-regulated in PD: third quadrant). A: Transcripts that did not reach statistical significance in the DTU analyses by either DRIMSeq or DEXSeq. B Transcripts found to be significant by both tools. C: Transcripts found to be significant by DEXSeq only. D: Transcripts found to be significant by DRIMSeq only. Transcripts identified as DTU events (significant after p-value adjustment) are coloured according to the plot legend. Red: transcript identified as a DTU event by both tools, yellow: transcript identified as a DTU event by DRIMSeq only, grey: transcript either didn't survive FWER correction by neither tool or wasn't nominally significant beforehand. (Transcripts can appear significant after FWER control even if they weren't nominally significant, due to StageR assigning significance by relying on the assumption that if DTU is occurring in the gene (that is: the gene has passed the screening stage) and one of its transcripts is significant, the other must subsequently also take part in the DTU to compensate).
(TIFF)

**S3 Fig. Overlap DTU genes and events, with and without cell correction.** DTU genes (A, B) and events (C, D) resulting from the analysis which included cell type estimations (purple) are overlapped with the results of the analysis where differences in cell types were not taken into account (turquoise). Only DTU events which were identified in the discovery cohort and replicated in the independent replication cohort were considered for this plot. A: DTU genes identified by DRIMSeq. B: DTU genes identified by DEXSeq. C: DTU events identified by DRIMSeq. D: DTU events identified by DEXSeq.
(TIFF)

**S4 Fig. Characteristics of the replicated DTU genes and events depicted as heatmaps.** Replicated DTU events (significant after OFWER correction in the discovery cohort, agreeing on the direction of change across cohorts and nominally significant at alpha = 0.05 in the replication cohort) are arranged in the y-axis. A: transcript's adjusted p-value (white cells indicate adjusted p-value >= 0.05). B: Transcript's log fold change (white cells correspond to transcripts not identified as DTU events). C: Transcript's nominal (uncorrected) p-value. In all heatmaps, characteristics are grouped by model design (i.e. with ("Incl. MGPs") or without ("w/o MGPs") accounting for MGPs) and by tool (DRIMseq or DEXSeq).
(TIFF)

**S5 Fig. Effect of pre-filtering on the number of transcripts per cohort.** A: Venn diagram for the sets of transcripts which survived pre-filtering in each cohort. Number of transcripts that survived filtering in the replication cohort (green), in the discovery cohort (red), and number of transcripts identified as DTU events in the discovery cohort (blue). B: Distribution of the correlation coefficients between transcript abundance (TPM) and sample RIN for non-concordant transcripts (i.e. transcripts removed during the pre-filtering in the replication cohort, but not in the discovery cohort) and concordant transcripts (i.e. transcripts that survived pre-filtering in both cohorts).
(TIFF)

**S1 File.** A: Cohort demographic and experimental information. B: DTU events. Table of identified DTU events, grouped by cohort (replication, discovery) and tool (DRIMSeq, DEXSeq). Gene-level and transcript-level p-values as reported by stageR (after FWER correction). Effect size corresponds to the coefficient of the condition variable (Control, PD) in the analysis model.
(XLSX)

**S1 Table. Overrepresentation analysis of DTU events in transcript biotypes.** P-values and odds ratios were determined by Fisher's exact test. The contingency table was built up separating transcripts by whether or not they were identified as DTU events and whether they were defined as the biotype of interest (as defined by Ensembl version 75). The rows are grouped by the tool which identified the DTU event and sorted by increasing p-value of the Fisher's exact test.
(PDF)

## Acknowledgments

We are grateful to patients and their families for participating in our research. We would also like to thank our colleagues at the Neuromics group for the fruitful discussions.

## Author Contributions

**Conceptualization:** Fiona Dick, Gonzalo S. Nido, Christian Dölle, Charalampos Tzoulis.

**Data curation:** Fiona Dick, Gonzalo S. Nido.

**Formal analysis:** Fiona Dick, Gonzalo S. Nido, Christian Dölle, Charalampos Tzoulis.

**Methodology:** Fiona Dick, Gonzalo S. Nido, Gry Hilde Nilsen, Christian Dölle.

**Project administration:** Charalampos Tzoulis.

**Resources:** Charalampos Tzoulis.

**Software:** Fiona Dick.

**Supervision:** Charalampos Tzoulis.

**Visualization:** Fiona Dick, Christian Dölle.

**Writing – original draft:** Fiona Dick.

**Writing – review & editing:** Fiona Dick, Gonzalo S. Nido, Guido Werner Alves, Ole-Bjørn Tysnes, Christian Dölle, Charalampos Tzoulis.

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
