## [Decision Letter · Decision Letter 0]

29 Jul 2020

Dear Dr Tzoulis,

Thank you very much for submitting your Research Article entitled 'Differential transcript usage in the Parkinson's disease brain' to PLOS Genetics. Your manuscript was fully evaluated at the editorial level and by three independent peer reviewers. The reviewers appreciated the attention to an important topic but identified several aspects of the manuscript that should be improved, with which the editors agree.

We therefore ask you to modify the manuscript to address the concerns of all three reviewers before we can consider your manuscript for acceptance. Your revisions should address each of the the specific points made by the reviewers. We also encourage following Reviewer 2's suggestion for including a diagram of ZNF189 to improve the clarity of Figure 2.

[LINK]

Yours sincerely,

Bruce A. Hamilton

Associate Editor

PLOS Genetics

Gregory Barsh

Editor-in-Chief

PLOS Genetics

Reviewer's Responses to Questions

**Comments to the Authors:**

Reviewer #1: I have uploaded comments as an attachment.

Reviewer #2: It’s concise and well-written, with a clear rationale both for the study and the methods used within the study.

From a technical standpoint, they’re using tools that model differential transcript usage (DTU) based on transcript counts. While there’s certainly an argument in the field as to how accurately these represent differential splicing, the authors are quite conservative in their methodology in that they: 1) follow a pipeline from experts in the field (https://bioconductor.riken.jp/packages/3.10/workflows/vignettes/rnaseqDTU/inst/doc/rnaseqDTU.html#stager-following-dexseq); 2) filter to remove lowly expressed transcript; 3) use two different methodologies (both DTU tools, but based on different underlying distributions) and 4) attempt to replicate in an external dataset. In other words, their methodology is robust.

Other methodological points that I think are worth mentioning/commending the authors for: 1) they correct for cell-type proportions, which is important given that the authors have shown in previous work (Nido et al. 2020) that this confounds gene expression; 2) they attempt to address how isoform changes might impact on disease by looking at effects on subcellular localisation, which I thought was imaginative; 3) the authors have made a very genuine effort to make their analyses transparent and the code reproducible. Looking through their GitHub repository, it is well documented and easy to navigate.

One major comment:

The authors have used GRCh37 and Ensembl v75; the latter was released in December 2013. Why have the authors not used GRCh38 and one of the later versions of Ensembl (which as of April 2020 is now at v100)?

A couple of minor comments:

First, the authors should be commended on their attempt to validate DTU with qPCR. However, I do have a few questions/comments.

The authors test ZNF189 arguing their choice is based on it having i) adequate individual transcript expression levels and ii) sufficiently distinct exonic composition of the individual transcripts to allow transcript-specific amplification. Could the authors expand on these criteria i.e. what was considered “adequate individual transcript expression” and “sufficiently distinct exonic composition”?

How many DTU genes would have passed the above criteria?

Further, it would be helpful for readers if authors provided a figure with the transcript structure of ZNF189 together with the primer sequences used for qPCR to demonstrate how primer choices would ensure transcript-specific amplification.

I assume that nominal significance is defined as nominal p-value < 0.05, but I could not find any definition of this in the manuscript. It would be helpful to add this.

I could not find any mention of what the read depth of the RNA-sequencing was. It would be useful to add this.

Reviewer #3: The authors present differential transcript usage in Parkinson’s disease (PD) and control brains in a discovery and replication cohort. The study presents novel results regarding the PD transcriptome and contributes to the understanding of disease mechanism. The bioinformatics analysis is rigorous, and the manuscript is well-written.

The main concern that I have is the lack of concordance between the discovery and replication cohorts. In the discovery cohort, 814 DTU events in 584 DTU genes were found, but only 23 DTU events in 19 genes were nominally significant and concordant in the replication cohort. This might be explained by the heterogeneity in brain tissue samples and the low number of samples used in the discovery and replication cohorts. The study results would be strengthened by follow-up qPCR analysis in both cohorts of a few of the key genes: THEM5, SLC16A1 and BCHE. Also, the lack of replication of previous findings (SNCA, PARK7, PARKN) should be discussed.

Some minor comments:

• The figures are low resolution and hard to read. It would be helpful to have the same order of transcript types for DEXSeq and DRIMSeq in Figure 1B.

• The authors state in methods that “The discovery cohort comprised individuals with idiopathic PD (n = 17) from the Park West study, a prospective population-based cohort which has been described in detail [15] and demographically matched controls (n = 11) from our brain bank for aging and neurodegeneration”. Does this mean that the PD and control samples were from different brain banks? I’m assuming that “our brain bank” is the same as Park West but the sentence is confusing and should be reworded.

**Have all data underlying the figures and results presented in the manuscript been provided?**

Reviewer #1: **No: **As detailed in my comment five, I think there are several data files that should be provided as supplemental files or in a Figshare repository to facilitate data sharing under FAIR principles.

Reviewer #2: Yes

Reviewer #3: Yes

PLOS authors have the option to publish the peer review history of their article (what does this mean?). If published, this will include your full peer review and any attached files.

Reviewer #1: **Yes: **Christopher DeBoever

Reviewer #2: No

Reviewer #3: No

---

## [Decision Letter · Decision Letter 1]

8 Oct 2020

Dear Dr Tzoulis,

We are pleased to inform you that your manuscript entitled "Differential transcript usage in the Parkinson's disease brain" has been editorially accepted for publication in PLOS Genetics. Congratulations!

Yours sincerely,

Bruce A. Hamilton

Associate Editor

PLOS Genetics

Gregory Barsh

Editor-in-Chief

PLOS Genetics

Comments from the reviewers (if applicable):

Reviewer's Responses to Questions

**Comments to the Authors:**

Reviewer #1: Thank you to the authors for addressing my comments. I have no further comments.

Reviewer #3: The authors have fully and very nicely addressed my comments.

**Have all data underlying the figures and results presented in the manuscript been provided?**

Reviewer #1: Yes

Reviewer #3: Yes

PLOS authors have the option to publish the peer review history of their article (what does this mean?). If published, this will include your full peer review and any attached files.

Reviewer #1: **Yes: **Christopher DeBoever

Reviewer #3: No

**Data Deposition**

http://datadryad.org/submit?journalID=pgenetics&manu=PGENETICS-D-20-00842R1

**Press Queries**

---

## [Editor Report · Acceptance letter]

23 Oct 2020

PGENETICS-D-20-00842R1 

Differential transcript usage in the Parkinson's disease brain 

Dear Dr Tzoulis, 

We are pleased to inform you that your manuscript entitled "Differential transcript usage in the Parkinson's disease brain" has been formally accepted for publication in PLOS Genetics! Your manuscript is now with our production department and you will be notified of the publication date in due course.

With kind regards,

Matt Lyles

PLOS Genetics

On behalf of:
